# Numerical Investigation of Terrain-Induced Turbulence in Complex Terrain Using High-Resolution Elevation Data and Surface Roughness Data Constructed with a Drone

**Takanori Uchida**

Research Institute for Applied Mechanics (RIAM), Kyushu University, 6-1 Kasuga-koen, Kasuga, Fukuoka 816-8580, Japan; takanori@riam.kyushu-u.ac.jp; Tel.: +81-92-583-7776; Fax: +81-92-583-7779

**Abstract:** Using the method based on unmanned aerial vehicle (UAV) imagery, two kinds of data can be obtained: the digital elevation model (DEM) for the digital expression of terrain, and the digital surface model (DSM) for the digital expression of the surface of the ground, including trees. In this research, a 3D topography model with a horizontal spatial resolution of 1 m was reproduced using DEM. In addition, using the differences between the DEM and DSM data, we were able to obtain further detailed information, such as the heights of trees covering the surface of the ground and their spatial distribution. Therefore, the surface roughness model and the UAV imagery data were directly linked. Based on the above data as input data, a high-resolution 3D numerical flow simulation was conducted. By using the numerical results obtained, we discussed the effect of the existence of surface roughness on the wind speed at the height of the hub of the wind turbine. We also discussed the effect of the differences in the spatial resolution in the horizontal direction of the computational grid on the reproductive precision of terrain-induced turbulence. As a result, the existence and the vortex structure of terrain-induced turbulence occurring near the target wind turbine was clearly revealed. It was shown that a horizontal grid resolution of about 5 m was required to reproduce terrain-induced turbulence formed from topography with an altitude of about 127 m. By the simulation using the surface roughness model, turbulence intensity higher than class A in the International Electrotechnical Commission (IEC) turbulence category was confirmed at the present study site, as well as the measured data.

**Keywords:** large-eddy simulation (LES); terrain-induced turbulence; complex terrain; drone

## 1. Introduction

Wind turbines are generally designed in compliance with power laws, in which wind speed gradually increases vertically upward and flows along the blades of the turbine. Therefore, when an irregular wind that deviates considerably from the power law passes through the wind turbines, the power-generating capacity significantly decreases and, in some cases, the wind load undergoes mechanical fatigue. This may lead to serious damage to its major parts, such as the yaw motor and yaw gears in the wind turbine nacelle. To reproduce invisible and complicated three-dimensional (3D) airflow structures that occur in the neighborhood of the wind turbines and visually grasp its flow and pattern, an approach based on computational fluid dynamics (CFD), such as Reynolds-Averaged Navier–Stokes (RANS) turbulence models and large-eddy simulation (LES) turbulence models, is useful [1–10].

Previous research showed that the irregular winds described above are due to terrain-induced turbulence caused by slight ups and downs in the terrain in the neighborhood of the wind turbine [11–14]. To precisely reproduce the terrain-induced turbulence on a computer via numerical simulation of the

wind conditions, accurately reproducing the terrain condition on site is crucial, including the degrees of ups and downs, the heights of trees covering the surface of terrain, and their spatial distribution.

Therefore, in this research, we reproduced the target wind turbine site using a 3D topography model constructed by photographs taken from an unmanned aerial vehicle (UAV, commonly known as a drone) with a spatial resolution of 1 m in the horizontal direction [15,16] and, by using it as input data into an LES turbulent model, we conducted a high-resolution 3D numerical flow simulation. Two types of data—a digital elevation model (DEM), as the digital expression of topography, and digital surface model (DSM), as the digital expression of ground surface, including trees—can be obtained by the method used in this research, based on UAV imagery. Therefore, precise information can be obtained of the heights of trees covering the surface of terrain and their spatial distribution by using the differences between both DEM and DSM data. In this research, we investigated the effect of the existence of surface roughness on wind speed distributions at the hub height of the wind turbine. The effect of the differences in spatial resolution in the horizontal direction of the computational grid on the reproduction accuracy of terrain-induced turbulence was also discussed. In this research, targeting the Atsumi Wind Farm constructed in Aichi Prefecture, Japan, where four large-sized commercial wind turbines with an output power of 2 MW (V80-2 MW, Vestas Wind System A/S) are installed, the above-mentioned methods were verified [13].

## 2. Materials and Methods

### 2.1. Summary of the Atsumi Wind Farm

This research was conducted targeting the Atsumi Wind Farm in Aichi Prefecture, Japan, which started operation in March 2007 (Figure 1). At Amami Wind Farm, four Vestas 2 MW (V80-2MW) large-sized commercial wind turbines are installed. The height of the center of the wind turbine hub from the ground was 78 m and the diameter of turbine blade is 80 m. The highest point of the blade tip was 118 m above ground (Figure 2). A wind direction sensor and an anemometer were installed on the nacelle of the wind turbine, and the information from these sensors was used for operation control of the wind turbine.

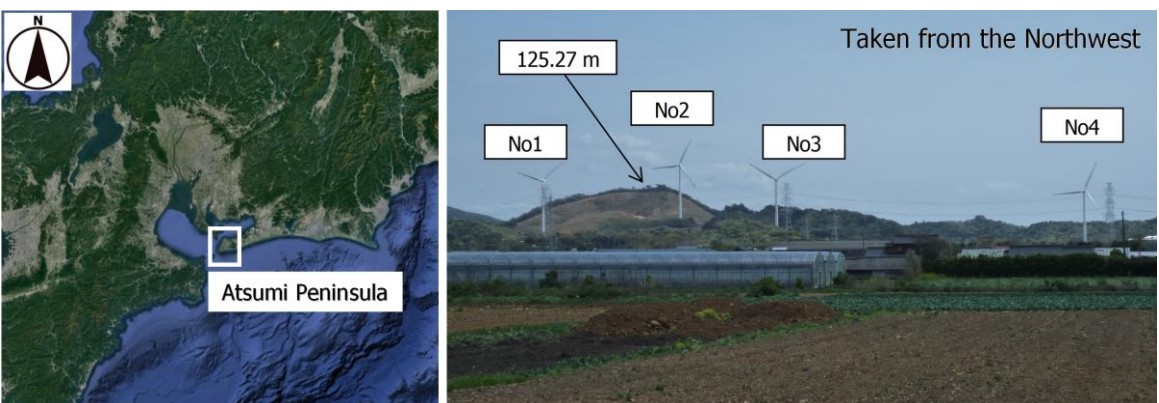

**Figure 1.** Four wind turbines at Atsumi Wind Farm, Aichi Prefecture, Japan.

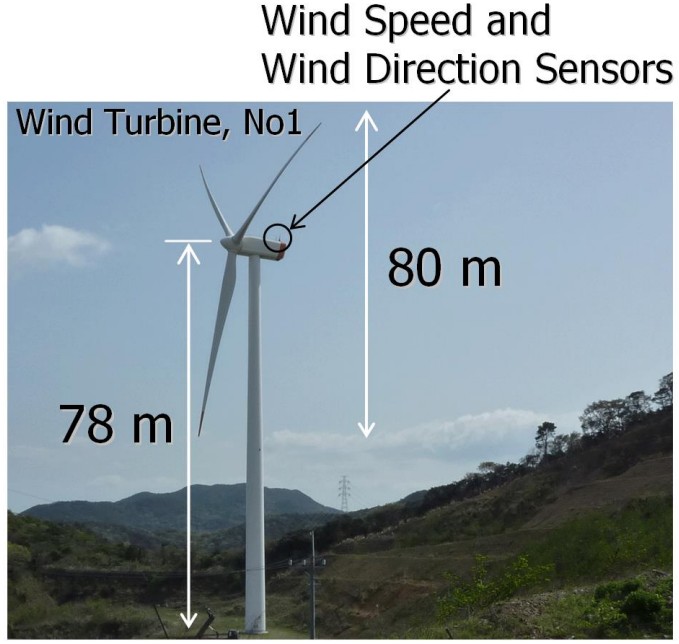

**Figure 2.** Photo of the No. 1 wind turbine showing specifications of the wind turbines.

*2.2. Analysis of Nacelle Propeller-Vane Anemometer Data (In-Situ Data Analysis)*

Figure 3 shows the wind rise that was calculated based on the output data of the wind direction sensor and anemometer installed on the nacelle of the No. 2 wind turbine targeted for this research. As a result, we found that the prevailing wind direction in this area is north-northwest. We also confirmed that east-southeast wind and southeast wind frequently occur. According to the report from a field office, when east-southeast winds occur, terrain-induced turbulence occurs, and, as a result, many difficulties occur in the main parts of the No. 2 wind turbine: the brake pad of yaw motor wears due to the high frequency of yaw control and the planetary gear fractures presumed to be due to metallic fatigue. Problems occur with the hydraulic operating device caused by frequent pitch control operations.

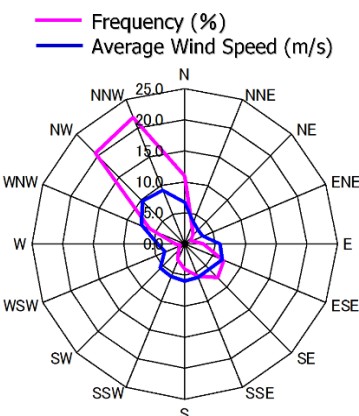

**Figure 3.** Wind rise diagram from April 2016 to March 2017 based on wind speed and wind direction sensors of the No. 2 wind turbine.

Figure 4 shows the distribution chart of turbulence intensity as function of wind speed, calculated based on the data output from the wind direction sensor and anemometer installed on the nacelle of the No. 2 wind turbine. As a result, in the case of north-northwest wind, which is the prevailing wind direction in this area, the turbulence intensity causing the wind turbine problem was

not large; however, in the case of east-southeast wind, the 90th percentile of turbulence intensity as a function of wind speed was large, above the class A value in the normal turbulence model (NTM) of the International Electrotechnical Commission (IEC). A series of results supports that under east-southeast, the No. 2 wind turbine is directly affected by terrain-induced turbulence. Therefore, in this research, we accurately reproduced the status of how the wind passes through the No. 2 wind turbine when the wind is east-southeast using high-resolution numerical wind condition simulation and discussed in detail the mechanism of the generation of terrain-induced turbulence. We also discussed the effects of the condition of trees covering the surface of terrain and the difference in the spatial resolution in the horizontal direction of the computational grid on the reproduction accuracy of terrain-induced turbulence and the effect of the height of the wind turbine hub on airflow characteristics.

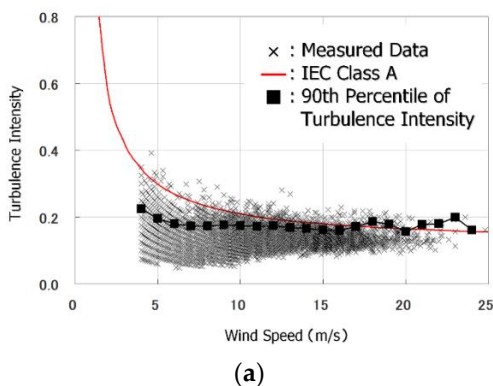 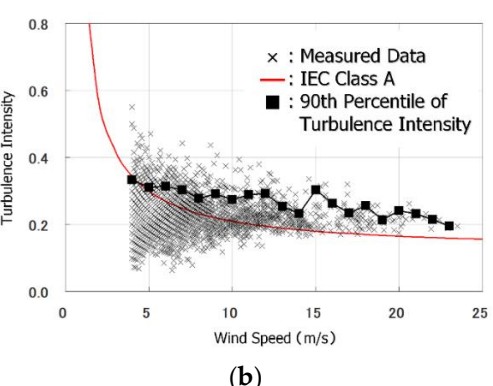

(**a**)                                        (**b**)

**Figure 4.** Turbulence intensity (TI) as a function of the 10 min mean wind speed of the No. 2 wind turbine from April 2016 to March 2017 when the wind was (**a**) NNW (north-northeast wind) and (**b**) ESE (east-southeast wind).

### 2.3. 3-D High-Resolution Topography Model Constructed from Drone Photographs

As a result of the analysis of turbulence intensity calculated from the measured data, we found that when the wind was east-southeast, the No. 2 wind turbine was directly affected by terrain-induced turbulence. In this research, to investigate in detail the mechanism generating terrain-induced turbulence, a survey by aerial photographs was conducted by a UAV, and we attempted to precisely reproduce the ups and downs of the terrain, the height of the trees covering the surface of the terrain, and their spatial distribution. The drone used in this research was a Phantom4 (Chinese drone manufacturer DJI) (Figure 5).

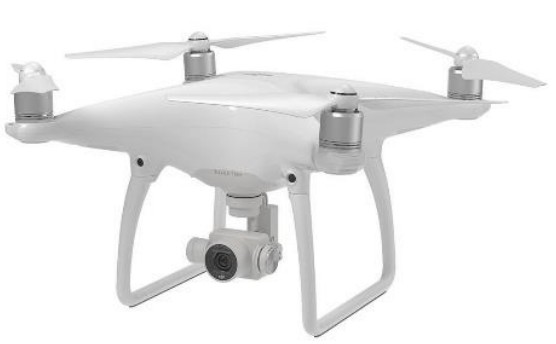 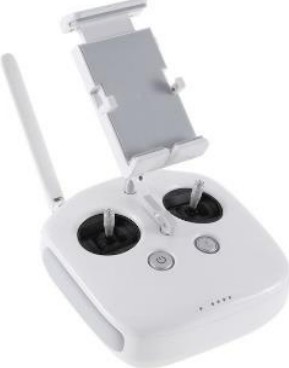

**Figure 5.** Unmanned aerial vehicle (UAV, commonly known as a drone) used in the present study (Phantom4, DJI).

As shown in Figure 6, the drone automatically flew all over the targeted area in a grid pattern and a 4 K camera was mounted on it to continuously capture bird's-eye photos from approximately

75 m above the ground. A group of approximately 570 images continuously recorded was processed with 3D image analyzing software (Drone2Map for ArcGIS), and 3D point cloud data were created. Thus, the obtained 3D point cloud data could not be directly used because errors would occur in the absolute accuracy of the position. Therefore, the coordinate was corrected using a 5 m mesh altitude provided by the Geospatial Information Authority of Japan. The images were divided into two groups, (1) geographic altitude data DEM and (2) data of tree heights covering the surface of the terrain (surface roughness data), and were output as mesh data (area 4 km², spatial resolution of 1 m in the horizontal direction) (Figure 7). DEM was structured by superimposing the detailed data of the area where the aerial measurements were recorded by a drone upon the 5 m mesh altitude provided by the Geospatial Information Authority of Japan. Only the surface roughness data in the area where the aerial measurement was conducted by a drone were reproduced, and other areas were regarded as smooth surfaces (no ground roughness). We conducted a series of data processing and data visualization using an ArcGIS Pro, Esri's next-generation 64-bit desktop geographic information system (GIS) software.

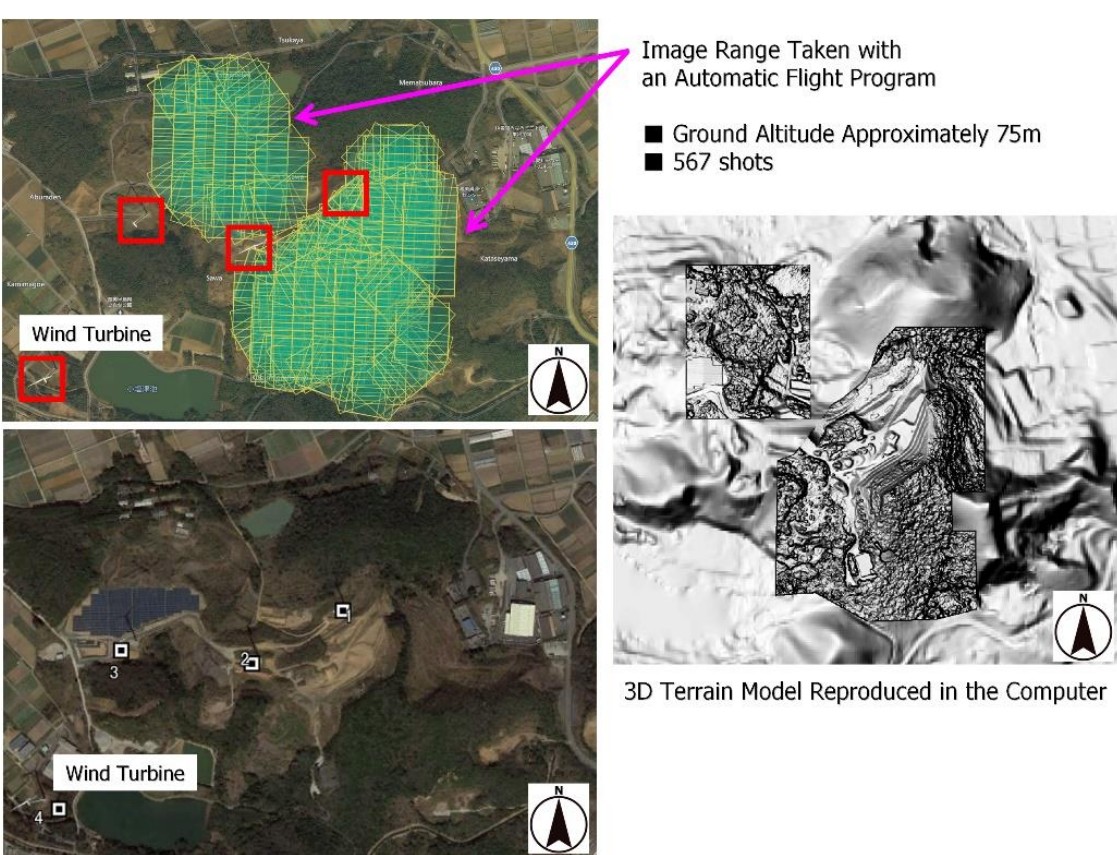

**Figure 6.** Construction of high-resolution elevation data with a drone.

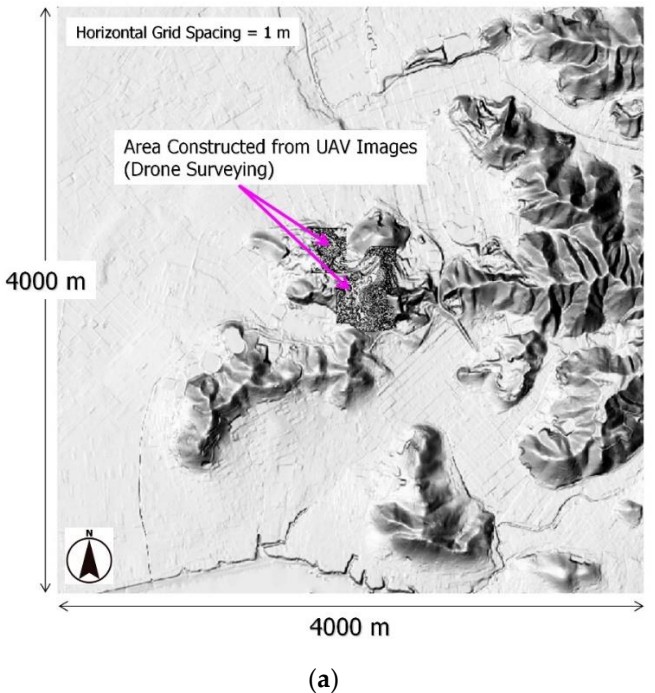

(**a**)

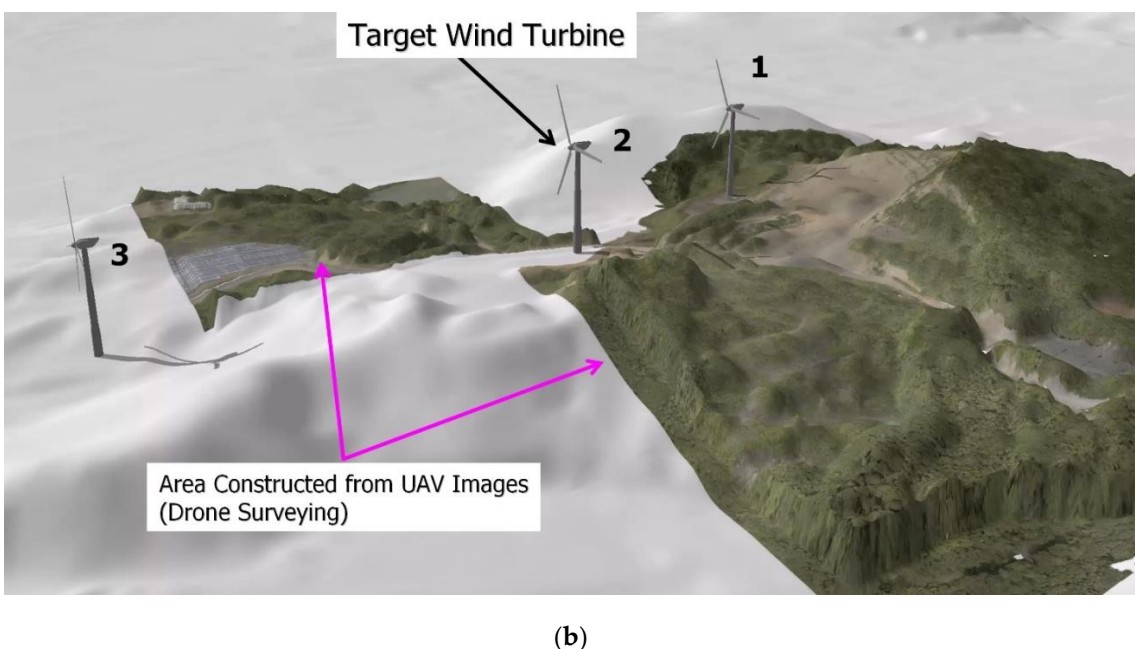

(**b**)

**Figure 7.** Three-dimensional topography model constructed by photographs taken from UAV: (**a**) top view and (**b**) bird's-eye view with surface roughness.

*2.4. Numerical Investigation into Terrain-Induced Turbulence Using LES Techniques*

In this research, we used a real terrain version RIAM-COMPACT (Research Institute for Applied Mechanics, Kyushu University, Computational Prediction of Airflow over Complex Terrain) based on the collocated grid of a general curvilinear coordinate system. The collocated grid system is characterized by functions to define physical velocity components and pressure at the cell center of computational grids and to define variables at the cell faces, which are obtained with a contravariant velocity component multiplied by the Jacobian determinant. The numerical calculation method is based on the finite-difference method (FDM) and incorporates a large-eddy simulation (LES) into the turbulence model. In LES, a spatial filter is set up in the flow field, and large- and small-sized turbulent

eddies are separated into those with a grid scale (GS) component larger than the computational grid and those with a sub-grid scale (SGS) component, which is smaller. The turbulent eddies with large GS components are directly applied in numerical simulation, independent of the model. The energy dissipative action due to the turbulent eddies with a small GS component is modeled based on physical consideration of SGS stress.

We applied a continuous equation of incompressible fluid with a filter operation (Equation (1)) and the Navier–Stokes equation (Equation (2)) in the dominant flow equations. The computational algorithm was based on the fractional step (F-S) method [17], and the time marching method was based upon Euler's explicit scheme. Poisson's equation of pressure was solved by successive over-relaxation (SOR). The second-order accurate central-difference scheme was used for the discretization of all spatial terms, excluding the convective term of Equation (2); the third-order upwind differencing scheme was used for convective terms. An interpolation method [18] was used for the fourth-order accurate central-difference scheme, constituting convective terms. The weight of the numerical dispersion terms of the third-order upwind differencing scheme was presumed to be $\alpha = 0.3$ against the $\alpha = 0.5$ of Kawamura–Kuwahara scheme type [19], and its effect was sufficiently small on the numerical results. A standard Smagorinsky model combined with a wall damping function [20] was used for the sub-grid scale model of LES and its model coefficient was 0.1 (Equations (3)–(8)). The effect of surface roughness was implemented as an external force using Equations (9) and (10). The surface roughness model used in this study was a simplified version of the canopy model described in the literature [21]. We used 5 and 10 for resistance coefficients (Cd) [22,23].

In the present study, the LES was assumed to reproduce the wind tunnel testing. Therefore, the effects of atmospheric stability associated with vertical thermal stratification of the atmosphere and inflow turbulence were neglected. The RANS results and the present LES results were compared in the latest findings [24] and the prediction accuracy of the present LES approach by comparison with wind tunnel experiments was discussed [25].

$$\frac{\partial \overline{u}_i}{\partial x_i} = 0 \tag{1}$$

$$\frac{\partial \overline{u}_i}{\partial t} + \overline{u}_j \frac{\partial \overline{u}_i}{\partial x_j} = -\frac{\partial \overline{p}}{\partial x_i} + \frac{1}{Re}\frac{\partial^2 \overline{u}_i}{\partial x_j \partial x_j} - \frac{\partial \tau_{ij}}{\partial x_j} - F_i \tag{2}$$

$$\tau_{ij} \approx \overline{u'_i u'_j} \approx \frac{1}{3}\overline{u'_k u'_k}\delta_{ij} - 2\nu_{SGS}\overline{S}_{ij} \tag{3}$$

$$\nu_{SGS} = (C_s f_s \Delta)^2 |\overline{S}| \tag{4}$$

$$|\overline{S}| = \left(2\overline{S}_{ij}\overline{S}_{ij}\right)^{1/2} \tag{5}$$

$$\overline{S}_{ij} = \frac{1}{2}\left(\frac{\partial \overline{u}_i}{\partial x_j} + \frac{\partial \overline{u}_j}{\partial x_i}\right) \tag{6}$$

$$f_s = 1 - \exp\left(-z^+/25\right) \tag{7}$$

$$\Delta = \left(h_x h_y h_z\right)^{1/3} \tag{8}$$

$$F_i = Cd \cdot \overline{u}_i \cdot V \tag{9}$$

$$V = \sqrt{\overline{u}^2 + \overline{v}^2 + \overline{w}^2} \tag{10}$$

*2.5. Summary of Simulation Parameters*

In this research, five numerical wind condition simulations were conducted (Table 1). The calculation area and computational grid in Case1 are shown in Figure 8a. The calculation

areas and computation grids from Case2_1 to Case4 are shown in Figure 8b. In Figure 8b, the areas where the surface roughness model was applied in Case3 and Case4 are shown in orange. In all cases, the calculation areas had a space of 2000 ($x$) × 1450 ($y$) × 900 ($z$) m in the streamwise direction ($x$), spanwise direction ($y$), and vertical direction ($z$). However, an artificial marginal area was set up on the upstream side of the calculation area and the ups and downs of the terrain were reduced to a flat area in the proportion of 95%. The maximum altitude was approximately 127 m and the minimum altitude was approximately 7 m in the calculation area. The geographical altitude data had a spatial resolution of 1 m, shown in Figure 7. The number of grids in the calculation areas, including marginal areas added to the upstream side and downstream sides, consisted of approximately 530,000 points, a total of 111 ($x$) × 59 ($y$) × 81 ($z$) points in each direction in Case1. For Case2_1 to Case4, the number of grids was approximately 10,000,000 points, a total of 451 ($x$) × 291 ($y$) × 81 ($z$) points in each direction. The vertical grid spacing in the $z$ direction decreased smoothly down to 0.35 m at the ground surface. The minimum grid width was the same in Case3 and Case4. Figure 9 shows the computational grids in the vertical direction near the terrain surface in Case3 and Case4. The area where the surface roughness model was applied included approximately 20 grid points in the $z$ direction.

**Table 1.** Simulation cases studied in the present study.

| | Case1 | Case2_1 | Case2_2 | Case3 | Case4 |
|---|---|---|---|---|---|
| Domain size (m) | | 2000 ($x$) × 1450 ($y$) × 900 ($z$) | | | |
| Horizontal grid resolution (m) | 25 | | 5 | | |
| Minimum vertical grid resolution (m) | | | 0.35 | | |
| Grid number | 111 ($x$) × 59 ($y$) × 81 ($z$) | | 451 ($x$) × 291 ($y$) × 81 ($z$) | | |
| Surface roughness model | | × | | ○ | |
| | | | | Cd = 5 | Cd = 10 |
| Non-dimensional time interval for estimating of turbulence statistics | 200 | | 600 | 200 | |

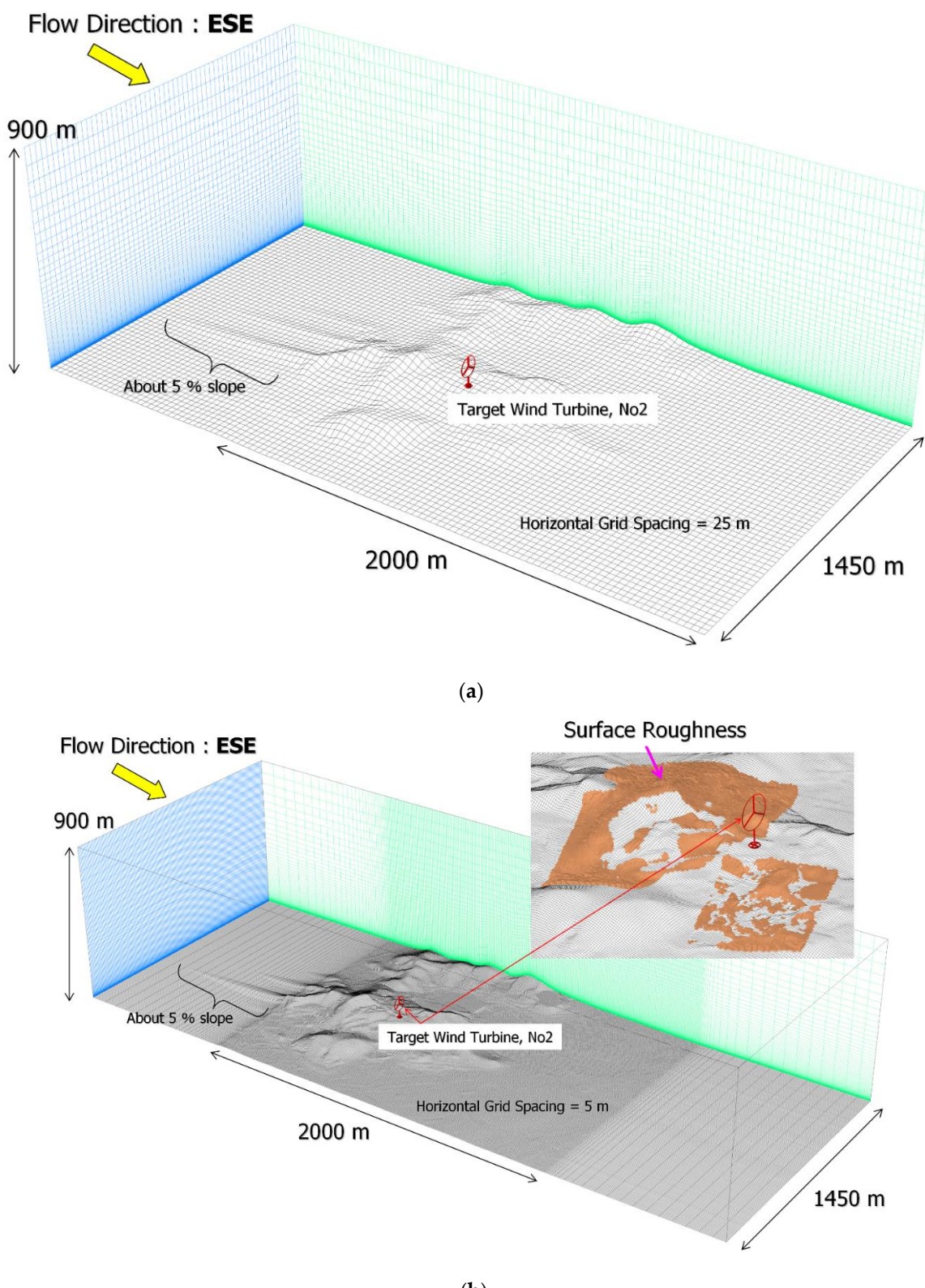

**Figure 8.** Computational grids for (**a**) Case1 and (**b**) Case2, Case3, and Case4.

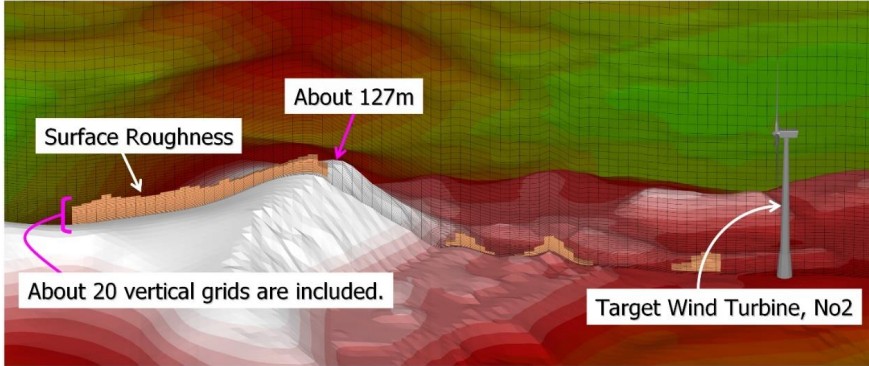

**Figure 9.** Vertical grids for Case3 and Case4 near the ground.

Concerning the boundary conditions, the power law distribution following $N = 7$, shown in Figure 10, was assigned to inflow conditions. In this research, to discuss the effects of terrain-induced turbulence originating from the ups and downs of the terrain, the inflow turbulence was omitted. Here, $z^*$ (m) of the vertical axis shows the height (m) from the surface of the terrain. The lateral interfaces and upper interface were assigned a slip condition and the outflow interface was assigned a convection-type outflow condition. A no-slip condition was assigned to the surface of the terrain. The non-dimensional parameter $Re$ in Equation (2) is a Reynolds number ($=U_{in} h/\nu$) and $Re = 10^4$. The handling of representative scales is shown in Figure 11, where $h$ represents the difference in elevation (120 m) within the calculation area, $U_{in}$ denotes the wind speed at the highest latitude position of the inflow boundary interface, and $\nu$ is the coefficient of kinematic viscosity. The time step was set to $\Delta t = 2 \times 10^{-3} \, h/U_{in}$. As previously explained, the numerical simulation in this research targeted the east-southeast wind for calculation.

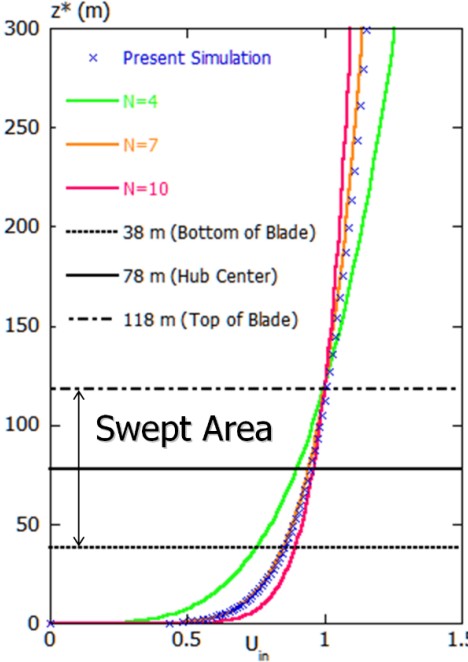

**Figure 10.** Inflow condition. $U_{in}$: the inflow streamwise wind velocity at the height of the maximum surface elevation in the computational domain; $z^*$ (m): the height above the ground.

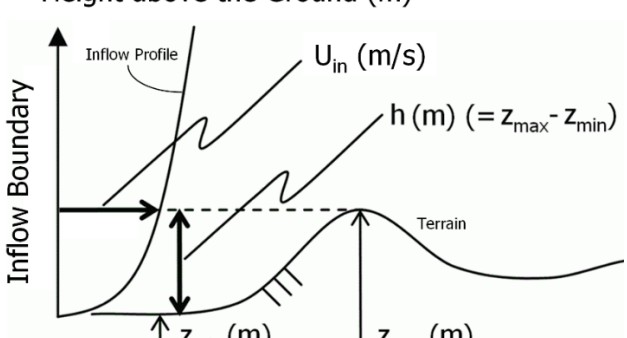

**Figure 11.** Two characteristic scales ($U_{in}$ and $h$).

## 3. Results and Discussion

Figure 12 shows the instantaneous flow field of wind speed distribution ($u$) in the streamwise direction ($x$) penetrating into the target wind turbine in Case1, Case 2_1, Case3, and Case4. Figure 13 shows the time-average flow field corresponding to Figure 12. The result of Case1 (Figure 12a) is different from those of Case2_1, Case3, and Case4, shown in Figure 12b–d. In Case1, the shedding of large-scale vortices from the hill at an altitude of approximately 127 m at the upper stream of the target wind turbine, which were observed in Case2_1, Case3, and Case4, was not reproduced. As a result, the laminarized flow was observed. This was considered to be caused by the grid resolution in the horizontal direction being as rough as 25 m in Case1, and therefore, the substantive value of Reynolds number was low. Consequently, when focusing on Figure 13a of the time-average flow field, a long, separated flow region in the $x$ direction was formed on the downstream of the hill located upstream of the target wind turbine.

In Case2_1, Case3, and Case4, where the horizontal grid resolutions were set to 5 m, as previously described, we observed the shedding of large-scale vortices from the hill at an altitude of approximately 127 m at the upper stream of the target wind turbine. Compared with Case1, the spatial resolution increased five times, and, as a result, the shedding of large-scale vortices from the hill at an altitude of approximately 127 m at the upper stream of the target wind turbine could be temporally and spatially resolved. As a result of the creation and detailed observation of the animation files, we found that the shedding of large-scale vortices produced from the hill at the upper stream of the target wind turbine and the generation and suppression of accompanying separated eddies were repeated at nearly regular intervals. Based on these results, the target wind turbine was visually observed to be directly influenced by terrain-induced turbulence. Through a precise comparison of the smooth surface of Case2_1, which did not have the surface roughness model applied, and Case3 and Case4, which did have the surface roughness model applied, and the coefficient parameters, which were changed from Cd = 5 to 10, we found that clear differences existed in the area influenced by the terrain-induced turbulence generated behind the hill. As shown by the down arrows in Figure 13b–d, the magnitude of the separated flow region in the $x$ direction, generated behind the hill at the upper stream of the target wind turbine, or in other words, the reattachment length of separated-shear layer from the hill, gradually extended in the $x$ direction in the order of Case2_1, Case3, and Case4.

Regarding the mechanism causing the difference in the reattachment length of the separated-shear layer, the following scenario can be expected: the wind speed deficit near the ground surface was greatly generated by surface roughness. The effect of surface roughness directly affected the vorticity generated from the surface of the hill upstream of the target wind turbine. In other words, the vorticity generated from the slope of the hill decreased and the reattaching strength of the separated-shear layer generated near the top of the hill weakened. As a result, compared with Case2_1 with a smooth surface without the surface roughness model applied, further elongation was shown in Case3 and Case4. With further elongation of the reattachment length of the separated-shear in the $x$ direction,

the area influenced by terrain-induced turbulence in the vertical direction ($z$) increased, as shown by the double arrows in Figure 13b–d. As a result, compared with Case2_1 with a smooth surface without the surface roughness model applied, the influence of terrain-induced turbulence on the target wind turbine was expected to increase in Case3 and Case4, which had the surface roughness model applied.

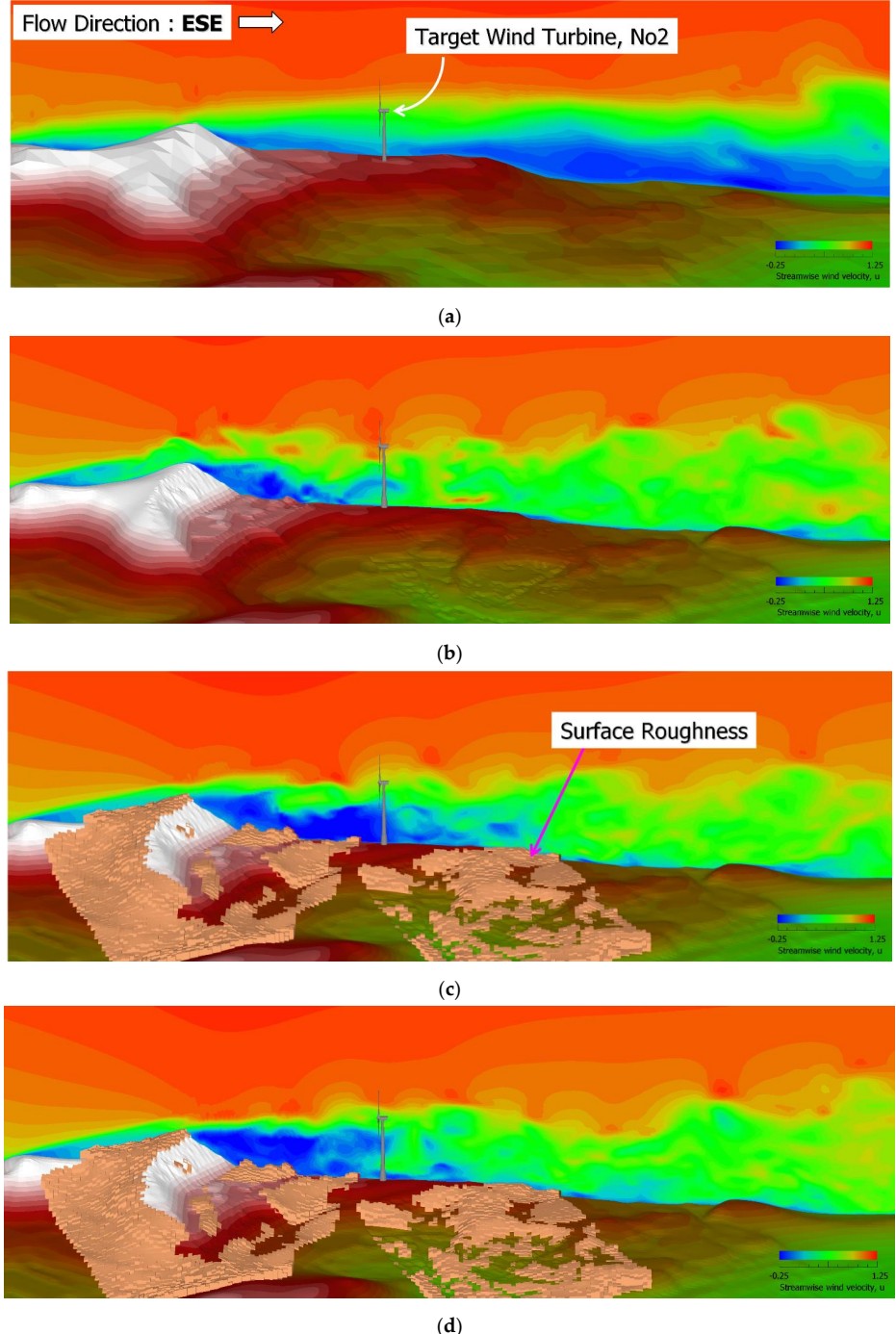

**Figure 12.** Instantaneous flow field at the wind turbine location for: (**a**) Case1, (**b**) Case2_1, (**c**) Case3, and (**d**) Case4. The effects of the wind turbines were not included in the simulations.

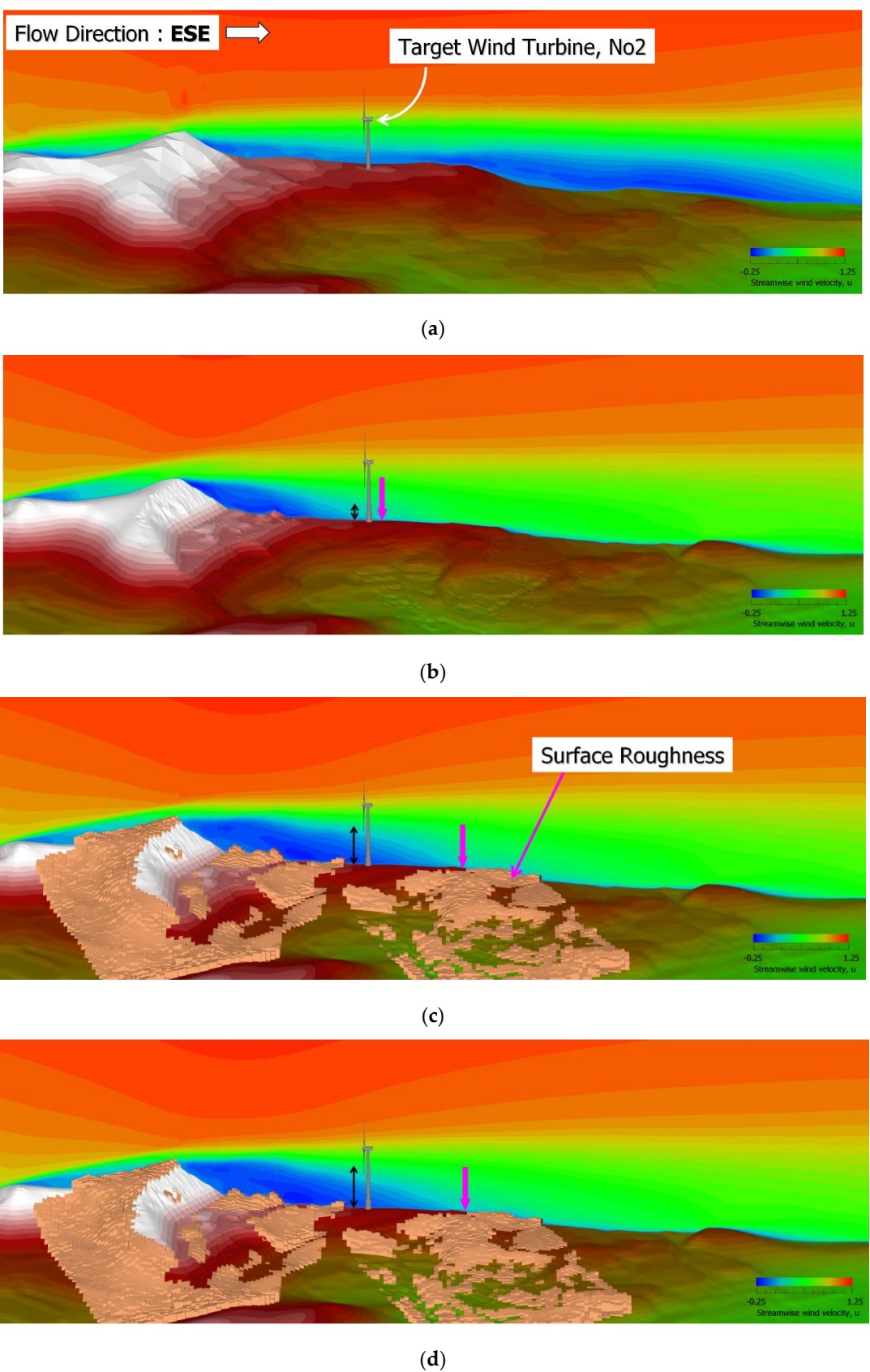

**Figure 13.** Time-averaged flow field at the wind turbine location for: (**a**) Case1, (**b**) Case2_1, (**c**) Case3, and (**d**) Case4. The effects of the wind turbines were not included in the simulations.

Figure 14 shows the temporal change in wind speed distribution ($u$) in the streamwise direction ($x$) at the height of the hub of the target wind turbine (78 m above the ground surface). Figure 14a compares Case2_1 and Case1, Figure 14b compares Case2_1 and Case3, and Figure 14c compares Case2_1 and Case4. When examining the comparison of Case2_1 and Case1 shown in Figure 14a, as previously mentioned, we found that the time histories of waveforms of both cases were significantly different. In Case2_1, with a grid resolution of 5 m in the horizontal direction, accompanied by the shedding of large-scale vortices from the hill at an altitude of approximately 127 m at the upper stream of the target wind turbine, the time history waveforms exhibited intense fluctuation in short cycles. These intensely fluctuating waveforms suggest that the gas flow field in the neighborhood of the target wind turbine had a complicated 3D structure. Conversely, in Case1, which had a grid resolution of 25 m in a horizontal direction, almost the same fluctuations were repeated in a long cycle. In other words, in Case1, a two-dimensional laminar flow was suggested to occur near the target wind turbine. From the comparison of Case2_1 and Case3 shown in Figure 14b, and that of Case2_1 and Case4 shown in Figure 14c, we observed similar tendencies.

This result clarified that due to the shedding of large-scale vortices originating from the hill upstream of the target wind turbine and the generation and suppression of accompanying separated eddies, a terrain-induced turbulence with a 3D structure was formed, and its influence directly extended to near the wind turbine.

Figure 15 shows the vertical distribution of the turbulence statistics at the location of the target wind turbine. Figure 15a shows the vertical distribution of mean wind speed. Figure 15b–d show the vertical distributions of the standard deviation in the streamwise direction ($x$), the spanwise direction ($y$), and the vertical direction ($z$), respectively. Here, $z^*$ of the vertical axis shows the height (m) from the surface of the ground. As mentioned, the result for Case1 tended to differ from those of Case2_1, Case3, and Case4 by being influenced by the difference in the flow field formed near the wind turbine. The distribution of standard deviation in each direction, in particular, became very small when the swept area of the wind turbine was $z^* = 38$–118 m, compared with that of Case2_1, Case3, and Case4. As shown in Figure 14a, this was the case because almost no temporal fluctuations occurred in the flow field formed near the wind turbine. Thus, regarding the comparison between Case2_1, Case3, and Case4, in the vertical distribution of the mean wind speed shown in Figure 15a, influenced by the differences in the influence range of terrain-induced turbulence from the hill, Case3 and Case4 showed almost the same trend; however, the differences in Case2_1 compared with Case3 and Case4 were confirmed. Then, each distribution of standard deviation in the $x$, $y$, and $z$ directions, shown in Figure 15b–d, respectively, were also discussed. Particular attention should be paid to the height from the ground, where the peak value of each standard deviation occurred, which is indicated by an arrow in these figures. Accompanied by the expansion of the influence range of terrain-induced turbulence in the $z$ direction, the peak value of each standard deviation occurred at a higher position in the $z$ direction in the order of Case2_1, Case3, and Case4. The Uchida–Kawashima scale_1 (U–K scale_1) is a turbulence evaluation index, defined in the literature [11]. The obtained value of U–K scale_1 exceeded the U–K scale_1 threshold of 0.2. From these results, it was revealed that the blades of the target wind turbine were directly and strongly affected by terrain-induced turbulence [11].

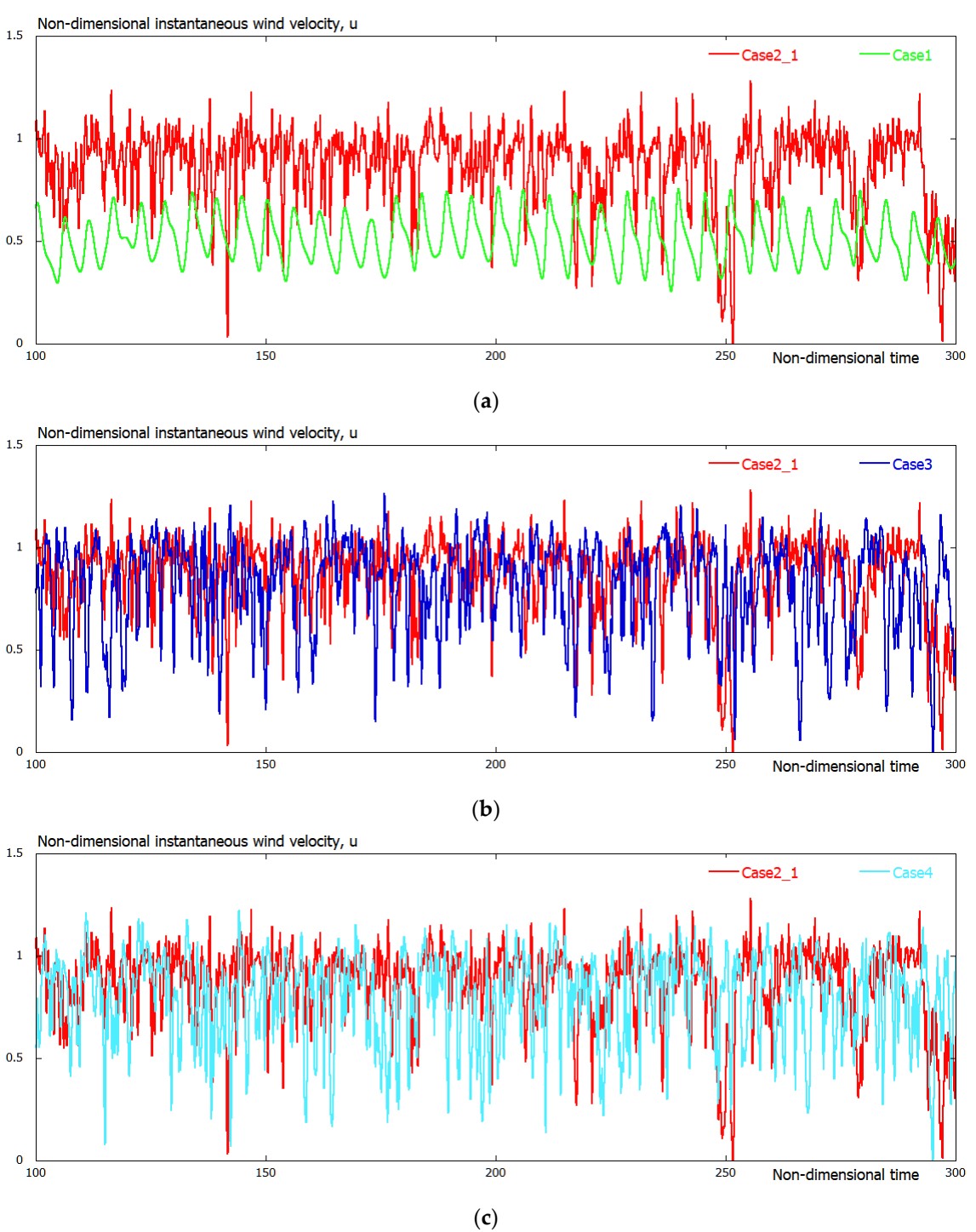

**Figure 14.** Comparison of temporal changes in the wind velocity component in the streamwise direction at the wind turbine hub height (78 m above the ground surface): (**a**) comparison of Case2_1 and Case1, (**b**) comparison of Case2_1 and Case3, and (**c**) Comparison of Case2_1 and Case4. The vertical and horizontal axes are normalized with the two characteristic scales ($U_{in}$ and $h$) shown in Figure 11.

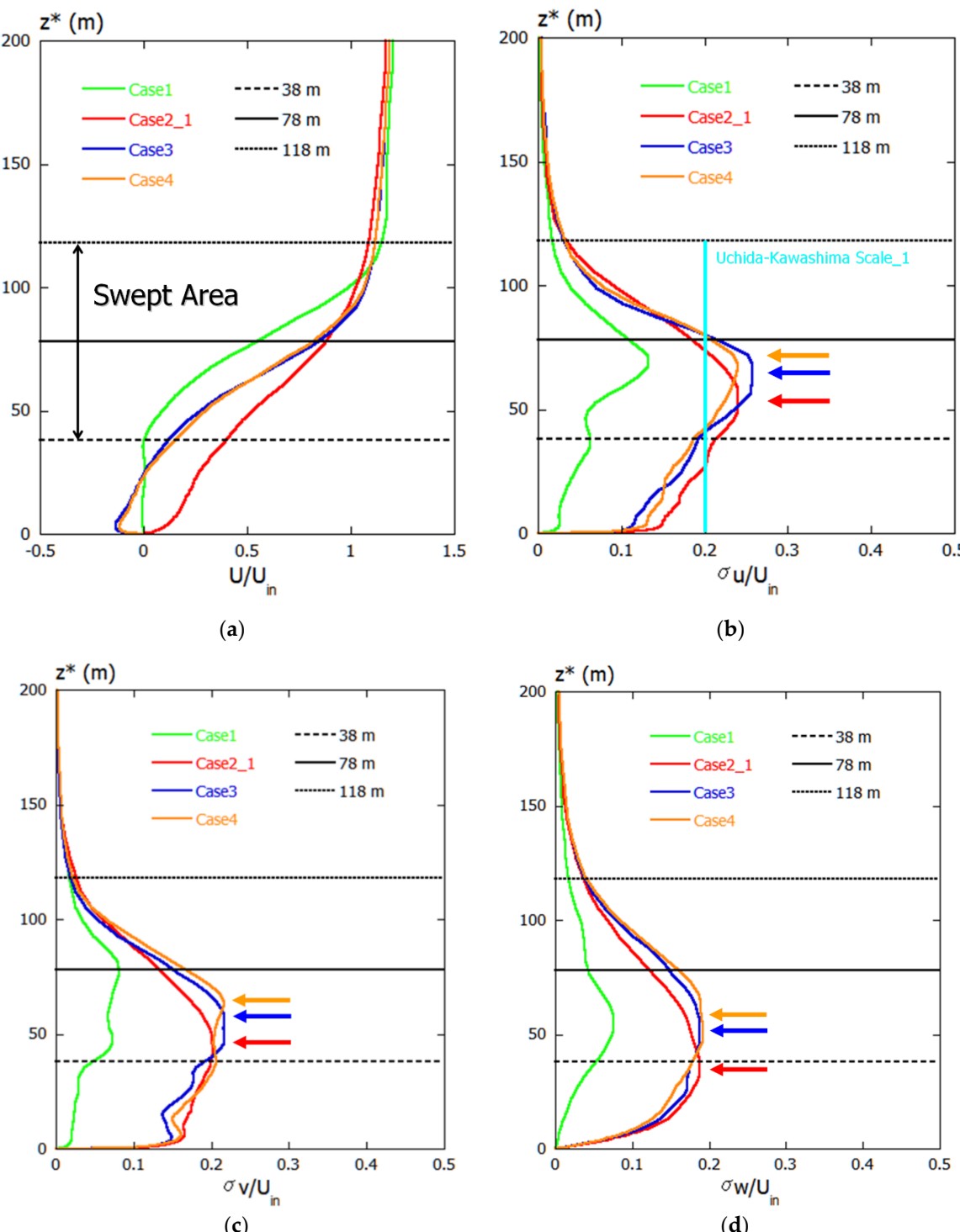

**Figure 15.** Vertical profiles of statistical quantities. (**a**) Normalized mean streamwise wind velocity, (**b**) normalized standard deviation in the streamwise direction, (**c**) normalized standard deviation in the spanwise direction, and (**d**) normalized standard deviation in the vertical direction. $U_{in}$: the inflow streamwise wind velocity at the height of the maximum surface elevation in the computational domain; $z^*$ (m): the height above the ground.

In Figure 16, Case2, with a smooth surface and without a surface roughness model, is used as an example, and the results of the discussion about a non-dimensional time interval for an estimation of turbulence statistics are shown. As shown in Table 1, the non-dimensional time interval of Case2_1 was

determined to be 200. In Case2_2, it was set to 600, which was three times that of Case2_1. No difference was observed in the vertical distribution of the mean wind speed and the vertical distributions of the standard deviation in the *x*, *y*, and *z* directions between Case2_1 and Case2_2. As a result, the validity of the non-dimensional time interval for estimating turbulence statistics was confirmed.

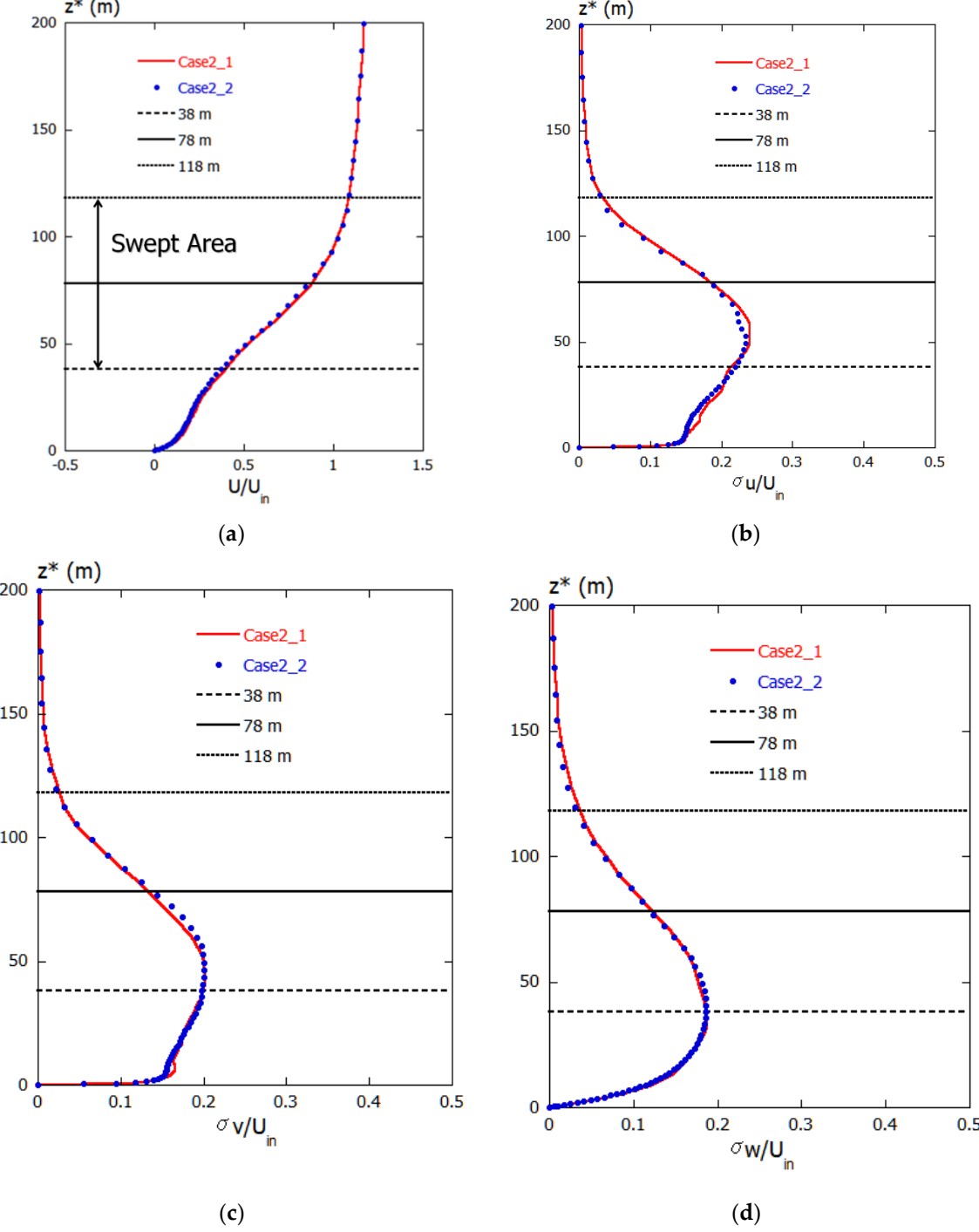

**Figure 16.** Comparison of vertical profiles of statistical quantities for Case2_1 and Case2_2. (**a**) Normalized mean streamwise wind velocity, (**b**) normalized standard deviation in the streamwise direction, (**c**) normalized standard deviation in the spanwise direction, and (**d**) normalized standard deviation in the vertical direction.

Finally, we estimated the turbulence intensity of a real-scale Case4 applied with the surface roughness model. Following the literature [12], the non-dimensional horizontal scalar wind velocity was defined from the wind speed component in the streamwise direction (*x*) and in the spanwise direction (*y*), which was then re-scaled so that its average value was 8 m/s on the real scale. The turbulence statistics used here were the data with a 200 time interval (non-dimensional time) used in Figure 15. Following the statistical processing method of general measured data, the time history waveform of the non-dimensional horizontal scalar wind velocity on the real scale obtained from this method was applied with a moving average interval of 10 min, and we calculated the average wind speed and the corresponding turbulence intensity. These results are shown in Figure 17. In this figure, the normal turbulence mode (NTM) defined in 614,001 Ed.3 (2005) of the IEC is also shown. Table 2 stipulates the classes of wind turbine.

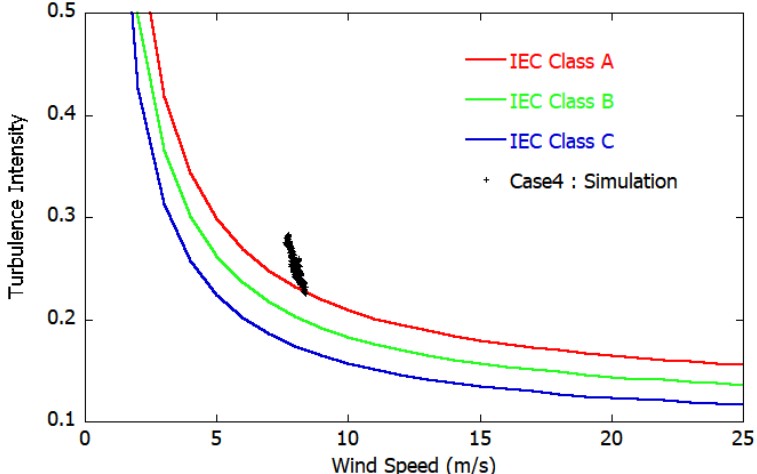

**Figure 17.** The relationship between turbulence intensity (TI) and wind speed at the wind turbine hub height (78 m above the ground surface) for Case4. Dots: simulation results from the present study. Lines: IEC (International Electrotechnical Commission) turbulence categories A, B, and C from the NTM (normal turbulence model), defined in IEC 61400-1 Ed.3 (2005).

**Table 2.** Wind turbine class by IEC 61400-1 Ed.3 (2005). $V_{ref}$ is the 50-year return period values with 10-min average wind speed; $I_{ref}$ is the expected value of TI for a wind speed of 15 m/s.

| | Class | A | B | C |
|---|---|---|---|---|
| | $V_{ref}$ | 50.0 | 42.5 | 37.5 |
| $I_{ref}$ | A | | 0.16 | |
| | B | | 0.14 | |
| | C | | 0.12 | |

NTM is defined as follows:

$$\sigma_{90q} = I_{ref}(0.75U + 5.6) \tag{11}$$

$$TI_{90q} = \frac{\sigma_{90q}}{U} = \frac{I_{ref}(0.75U + 5.6)}{U} \tag{12}$$

where *TI* is the turbulence intensity, *U* is the 10-min average streamwise wind velocity (m/s), $\sigma$ is the wind velocity standard deviation (m/s), and subscript $90q$ is the 0.9th quantile value.

Designers design wind turbines to satisfy the wind turbine classes and turbulence categories. Wind power operators can reduce business risk by confirming that the turbulence intensity at the site is at a lower turbulence intensity distribution than the envelope expressed using Equation (12). By focusing on Figure 17, the turbulence intensity reproduced by this method was above the limit

of the class A in the IEC turbulence category, similar to the measured data shown in Figure 4b. From the above, we demonstrated that it is possible to numerically reproduce turbulence intensity on complicated terrain.

## 4. Conclusions

Using the method based on UAV imagery, two kinds of data can be obtained: the DEM for the digital expression of terrain, and the DSM for the digital expression of the surface of the ground, including trees. In this research, a 3D topography model with a horizontal spatial resolution of 1 m was reproduced using DEM. In addition, using the differences between the DEM and DSM data, we were able to obtain further detailed information, such as the heights of trees covering the surface of the ground and their spatial distribution. Therefore, the surface roughness model and the UAV imagery data were directly linked. Based on the above data as input data, a high-resolution 3D numerical flow simulation was conducted. By using the numerical results obtained, we discussed the effect of the existence of surface roughness on the wind speed at the height of the hub of wind turbine. We also discussed the effect of the differences in the spatial resolution in the horizontal direction of the computational grid on the reproductive precision of terrain-induced turbulence. As a result, the existence and the vortex structure of terrain-induced turbulence occurring near the target wind turbine was clearly revealed. It was shown that a horizontal grid resolution of about 5 m was required to reproduce terrain-induced turbulence formed from topography with an altitude of about 127 m. By the simulation using the surface roughness model, turbulence intensity higher than class A in the IEC turbulence category was confirmed at the present study site as well as the measured data.

**Funding:** This research was supported by JSPS KAKENHI Grant Number 17H02053.

**Acknowledgments:** To conduct this research, the authors were provided with the various types of data on the Atsumi Wind Farm, Aichi Prefecture, Japan by Kyudenko Corporation. The authors would like to express their gratitude to all the organizations.

**Conflicts of Interest:** All authors declare no conflict of interest.

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
