# Peer review of "Numerical Investigation of Terrain-Induced Turbulence in Complex Terrain Using High-Resolution Elevation Data and Surface Roughness Data Constructed with a Drone"

_energies, doi:10.3390/en12193766_

Round 1

Reviewer 1 Report

The manuscript explores aspects of the simulation of the local wind field at a wind turbine site. The wind turbine site is strongly affected by the topography and surface vegetation, particularly for the specific turbine location studied because it is located just downwind of a relatively steep hill. The study discusses some of the challenges of such simulations, showing that the details of the topography and surface can be critical for accurate simulations. Also, some aspects of the model setup, such as grid resolution, can have a large impact on the results. It is shown that detailed topographical information gathered by a drone can help improve the quality of the simulations. The study is well designed and the manuscript is also structured well. The manuscript is fairly easy to follow (the clarity of some parts can be somewhat improved). The manuscript is overall interesting and, I believe, a welcome addition to the literature. The main weakness of the study it that the overall goal is not clear. Thus, even though there is a clear logical development of the concepts through the manuscript, the problem that is actually addressed is vague. This can be corrected by restructuring some of the introduction and conclusions, by being clearer throughout what are the study’s questions and aims, and by avoiding a couple of instances of unsupported inferences.

Specifically:

The title implies that the main focus of the study is a method that directly and analytically links high-resolution elevation data to the simulation of turbulence. However, the reader is rather disappointed to find out that the drone measurements are used because detailed information about the wind turbine site is not available. Moreover, the values of the drag coefficients, Cd, and the height of the application of the drag force, Fi in equation (1), are not linked to the drone data. The values and location seem rather ad hoc and empirical.

The simulations appear to imply that the most important parameter for an accurate simulation is the grid resolution. It is not clear if a simulation with 25 m horizontal resolution and the “surface roughness model” will produce accurate results. This could be an important conclusion: that a threshold in grid resolution exists before the “surface roughness model” effects become important. It is not clear if the resolution requirement is because of limitation of the turbulence parameterization or because of the specific topography.

There is no validation of the simulations. The manuscript implies that there are velocity data collected at the wind farm site. There are also statements such as “the IEC category was confirmed” but this was not stated as a question or goal of the study and appear disconnected from the rest of the discussion.

Last sentence: “we clarified that turbulence intensity can be numerically produced using the present method”. This is vague and confusing. Is the goal to reproduce some turbulence of unspecified intensity? Should we expect a specific value to be captured in the simulations. Is this one of the main goals of the study? How does it connect to the use of the drone data as stated in the title of the manuscript?

Other comments:

The drag model for vegetation is referred to as “surface roughness model”. In meteorology, the term “surface roughness” corresponds to unresolved roughness, that is, surface fluctuations less that about Dz/10. The drag model presently used is more similar to a plant canopy or vegetation model.

Cases 2_1 and 2_2 are essentially the same simulation, which usually is documented as one case in a table. The different time interval could be used without referring to a different case.

It should be clearly stated that the effects of the wind turbines are not included in the simulations and that the turbines are shown in figures 12 and 13 to denote their position.

There is a hypothesis regarding the variation of the reattachment length (page 11) but there is no literature cited to support this hypothesis. The statements are not clear: “the vorticity (momentum) generated from the slope of microtopography”.Vorticity and momentum are different quantities and it is not clear which is the main cause of the variation of the reattachment length. Also, the component of vorticity or momentum is not specified.

The term “microtopography” is used in many places. Perhaps it is better to just refer to the hill upstream of turbine No 2, rather than use a generic term which a specific topographical feature is discussed.

“background of the microtopography” should be “downstream of the hill”

“substantive value of Reynolds number” should be “effective value”? Note that the actual Reynolds number is really high.

In several places there is reference to the “mainstream direction (x)”. Should this be replaced with the more standard “streamwise direction”?

page 14: “turbulence was possibly considerably affecting the target wind turbine”. This is vague.

Author Response

I will reply to comments in a PDF file.

Reviewer 2 Report

Please see the attached report for the referee's comments.

Author Response

I will answer questions in a PDF file.

Round 2

Reviewer 1 Report

The author has satisfactorily addressed my concerns, most by modifications of the manuscript. 

Reviewer 2 Report

Author has addressed my comments and the manuscript in the present form can be accepted for publication.